# An LGAD-Based Full Active Target for the PIONEER Experiment

Simone Michele Mazza 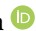

Santa Cruz Institute for Particle Physics (SCIPP), Unversity of California Santa Cruz, Santa Cruz, CA 95064, USA; simazza@ucsc.edu

**Abstract:** PIONEER is a next-generation experiment to measure the charged pion branching ratios to electrons vs. muons $R_{e}/\mu = \frac{\Gamma(\pi^+ \to e^+ \nu(\gamma))}{\Gamma(\pi^+ \to \mu^+ \nu(\gamma))}$ and pion beta decay (Pib) $\pi^+ \to \pi^0 e\nu$. The pion to muon decay ($\pi \to \mu \to e$) has four orders of magnitude higher probability than the pion to electron decay ($\pi \to e\nu$). To achieve the necessary branching-ratio precision it is crucial to suppress the $\pi \to \mu \to e$ energy spectrum that overlaps with the low energy tail of $\pi \to e\nu$. A high granularity active target (ATAR) is being designed to suppress the muon decay background sufficiently so that this tail can be directly measured. In addition, ATAR will provide detailed 4D tracking information to separate the energy deposits of the pion decay products in both position and time. This will suppress other significant systematic uncertainties (pulse pile-up, decay in flight of slow pions) to <0.01%, allowing the overall uncertainty in to be reduced to O (0.01%). The chosen technology for the ATAR is Low Gain Avalanche Detector (LGAD). These are thin silicon detectors (down to 50 μm in thickness or less) with moderate internal signal amplification and great time resolution. To achieve a 100% active region several emerging technologies are being evaluated, such as AC-LGADs and TI-LGADs. A dynamic range from MiP (positron) to several MeV (pion/muon) of deposited charge is expected, the detection and separation of close-by hits in such a wide dynamic range will be a main challenge. Furthermore, the compactness and the requirement of low inactive material of the ATAR present challenges for the readout system, forcing the amplifier chip and digitizer to be positioned away from the active region.

**Keywords:** pion decay; LGAD; timing detectors



## 1. Introduction

The branching ratio $R_{e/\mu} = \frac{\Gamma(\pi^+ \to e^+ \nu(\gamma))}{\Gamma(\pi^+ \to \mu^+ \nu(\gamma))}$ for pion decays to electrons over muons provides the best test of electron–muon universality in charged-current weak interactions. Furthermore, it is extremely sensitive to new physics at high mass scales. In the Standard Model (SM), $R_{e/\mu}$ has been calculated with extraordinary precision at the $10^{-4}$ level as [1–3]

$$R_{e/\mu} \text{ (SM)} = (1.2352 \pm 0.0002) \times 10^{-4}, \tag{1}$$

perhaps the most precisely calculated weak interaction observable involving quarks. Because the uncertainty of the SM calculation for $R_{e/\mu}$ is very small and the decay $\pi^+ \to e^+\nu$ is helicity-suppressed by the $V - A$ structure of charged currents, a measurement of $R_{e/\mu}$ is extremely sensitive to the presence of pseudo-scalar (and scalar) couplings absent from the SM; a disagreement with the theoretical expectation would unambiguously imply the existence of new physics beyond the SM. With 0.01% of experimental precision, new physics beyond the SM (BSM) up to the mass scale of 3000 TeV may be revealed by a deviation from the precise SM expectation [3]. Currently, the most accurate measurement was reported by PIENU [4],

$$R_{e/\mu} \text{ (Expt)} = (1.2344 \pm 0.0023\text{(stat)} \pm 0.0019\text{(syst)}) \times 10^{-4}, \tag{2}$$

at the 0.2% precision level. The result is in excellent agreement with the SM expectation in contrast to recent hints of violation of third-generation Lepton Flavor Universality (LFU) in some *B*-meson decays [5] as discussed below. TRIUMF PIENU [6,7] and PSI PEN [8–10] experiments expect to improve the measurement precision by another factor of 2 or more to a level of ≤0.1%. However, even when these goals are realized, this still leaves room for experimental improvement by more than an order of magnitude in uncertainty to compare with the SM prediction.

Precision measurements of beta decays of neutrons, nuclei, and mesons provide very accurate determinations of the elements $|V_{ud}|$ and $|V_{us}|$ of the CKM quark-mixing matrix [11,12]. Recent theoretical developments on radiative corrections and form factors have led to a $3\sigma$ tension with CKM unitarity which, if confirmed, would point to new physics in the multi-TeV scale (see, e.g., Ref. [13]). The detector optimized for a next-generation $R_{e/\mu}$ experiment will also be ideally suited for a high-precision measurement of pion beta decay and searches for exotic pion and muon decays. The branching ratio for pion beta decay was most accurately measured by the PiBeta experiment at PSI [14–18] to be

$$\frac{\Gamma(\pi^+ \to \pi^0 e^+ \nu)}{\Gamma(Total)} = 1.036 \pm 0.004(stat) \pm 0.004(syst) \pm 0.003(\pi \to e\nu) \times 10^{-8}, \quad (3)$$

where the first uncertainty is statistical, the second systematic, and the third is the $\pi \to e\nu$ branching ratio uncertainty. Pion beta decay, $\pi^+ \to \pi^0 e^+ \nu(\gamma)$, provides the theoretically cleanest determination of the CKM matrix element $V_{ud}$. With current input one obtains $V_{ud} = 0.9739(28)_{\exp}(1)_{\text{th}}$, where the experimental uncertainty comes almost entirely from the $\pi^+ \to \pi^0 e^+ \nu(\gamma)$ branching ratio (BRPB) [16] (the pion lifetime contributes $\delta V_{ud} = 0.0001$), and the theory uncertainty has been reduced from $(\delta V_{ud})_{\text{th}} = 0.0005$ [19–21] to $(\delta V_{ud})_{\text{th}} = 0.0001$ via a lattice QCD calculation of the radiative corrections [22]. The current precision of 0.3% on $V_{ud}$ makes $\pi^+ \to \pi^0 e^+ \nu(\gamma)$ not presently relevant for the CKM unitarity tests because super-allowed nuclear beta decays provide a nominal precision of 0.03%.

## 2. PIONEER

PIONEER is a next-generation experiment to measure $R_{e/\mu}$ and pion beta decay (Pib) with an order of magnitude improvements in precision from the PIENU, PEN, and PIBETA experiments. The experiment may be proposed to be initiated at one of the pion beam lines (either PiE1 or PiE5) at PSI in a timescale of 5 years. To achieve the necessary precision for both $R_{e/\mu}$ and Pib PIONEER has two main detectors: a high granularity full silicon active target (in short ATAR) and a 28 $X_0$ segmented calorimeter (liquid Xenon or LYSO) with high energy resolution. Furthermore, a one-layer tracker will be placed around the ATAR to tag exiting positrons. The ATAR, having a few mm of active thickness, would be sitting at the most probable pion interaction point and the calorimeter would be a sphere of a 1 m radius around it. A schematic of the PIONEER external enclosure can be seen in Figure 1, left. The ATAR region is highlighted in red in the shown schematic. The following sections will be focused on the ATAR detector development.

Two scintillation calorimeter options are presently under consideration: Liquid xenon [23] and LSO crystals [24]. Evaluations of potential energy and timing performance, rate capabilities, mechanical and cryogenic facilities, and cost are ongoing. Both technology candidates have high light output fast timing and near-complete containment of EM showers.

The ATAR would provide a high precision 4D tracking that would allow separating the energy deposits of the pion decay products in both position and time (as shown in Figure 1, Right). Furthermore, it will allow suppressing other significant systematic uncertainties (pulse pile-up, decay in flight of slow pions) to <0.01%. The combination of the high precision calorimeter and the ATAR will allow the overall uncertainty in the measurements of $R_{e/\mu}$ and Pib to be reduced to O (0.01%) and O (0.05%) respectively. For

the $R_{e/\mu}$ phase $3 \times 10^5$ $\pi^+$ per second (around $10^{12}$ total $\pi^+$ per year) are expected, for the Pib phase $1.5 \times 10^7$ $\pi^+$ per second (around $10^{14}$ total $\pi^+$ per year) are expected.

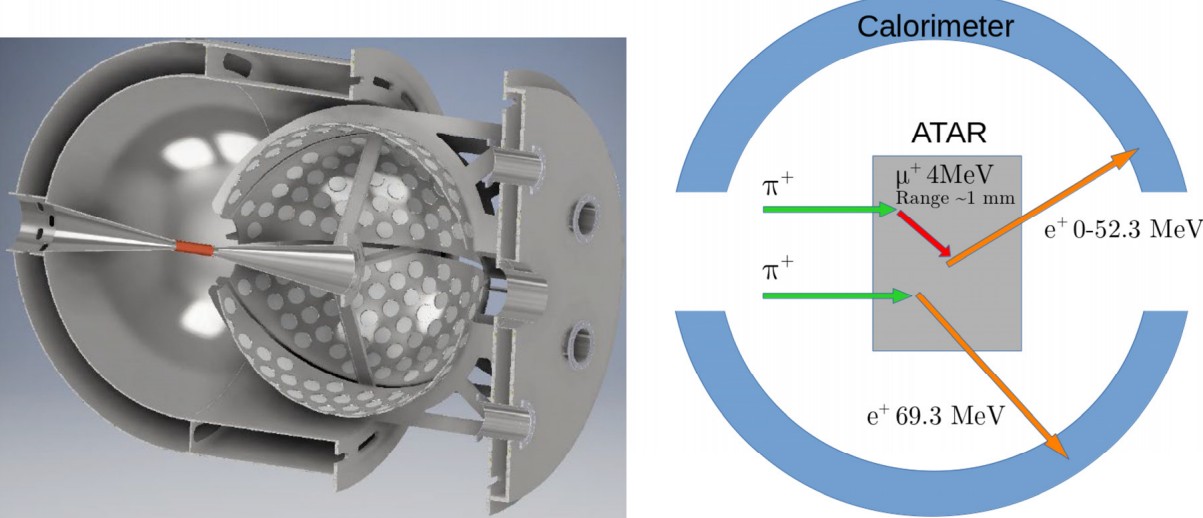

**Figure 1.** (**Left**) External enclosure for PIONEER calorimeter and ATAR. (**Right**) Pion decay to muon or electron in the ATAR and Calorimeter.

### 3. Active Target (ATAR)

A highly segmented active target (ATAR), Figure 2 (Left), is a key new feature of the proposed PIONEER experiment. It will define fiducial pion stops, suppress decay in flight, suppress the $\pi \to \mu \to e$ Michel chain, which is four orders of magnitude more likely than the rare $\pi_{2e}$ process $\pi \to e\nu$, and will furnish selective event triggers. Figure 2 (Right) shows the simulated pion and muon decay position in the ATAR. Ultimately, it is expected that ATAR will be able to suppress the $\pi \to \mu \to e$ chain by several orders of magnitude. Those events are identified using the $\sim$1.5 ns pulse pair resolution of the Low Gain Avalanche Detectors (LGADs) and a tight positron observation window of about one pion lifetime, which disfavors the slower muon decay. This would reduce the number of muon decay electrons sufficiently so that the low energy tail of the calorimeter $\pi_{2e}$ response extending below the maximum Michel energy can be directly measured. Furthermore, it will limit accidental muon stops that precede the trigger signal. These were a significant background in the previous generation of experiments and had to be suppressed by pile-up rejection at the expense of event rate. This can be achieved by checking whether the observed positron belongs to the pion stopping vertex using additional tracking detectors. The ATAR will also identify decays in flight. Muons arising from upstream pion decays are relatively easily identified by their energy loss properties and by kinks in their trajectories. Pion decay inside the target will be separated by kinks in the topology, dE/dx along the track, and range in the target.

To fulfill these goals the ATAR would need to be able to detect both the exiting $e^+$, a Minimum Ionizing Particle (MiP), and larger ($\sim$100 MiPs) energy deposits from $\pi^+$ and $\mu^+$. The large dynamic range, order of $\sim$2000, of the signals provides a significant challenge for the readout electronic both in the amplification and digitization stage.

The ATAR tentative design dimensions are $2 \times 2$ cm$^2$ transverse to the beam, in the beam direction individual silicon sensors are tightly stacked with a total thickness of roughly 6 mm. The requirement of avoiding dead material in the detector volume precludes the use of bump bonded electronics readout chips. For this reason, a strip geometry with the electronic readout connected on the side of the active region is foreseen. The strips are oriented at $90^0$ to each other in subsequent staggered planes to provide the measurement of both coordinates of interest and allow space for the readout and wirebonds. The detectors

are paired with the high-voltage facing each other in a pair to avoid ground and high voltage in proximity. In the design, the strips are wire bonded to a flex, alternating the connection on the four sides, which brings the signal to a readout chip positioned a few cm away from the active volume. This brings the chip outside of the path of the exiting positrons, reducing the degradation of their energy resolution. The readout electronics sit on a PCB connected to the first flex, then a second flex, which also provides the various voltages and grounds, bringing the amplified signal to the digitizers in the back-end. A schematic of the ATAR is shown in Figure 3 (Left).

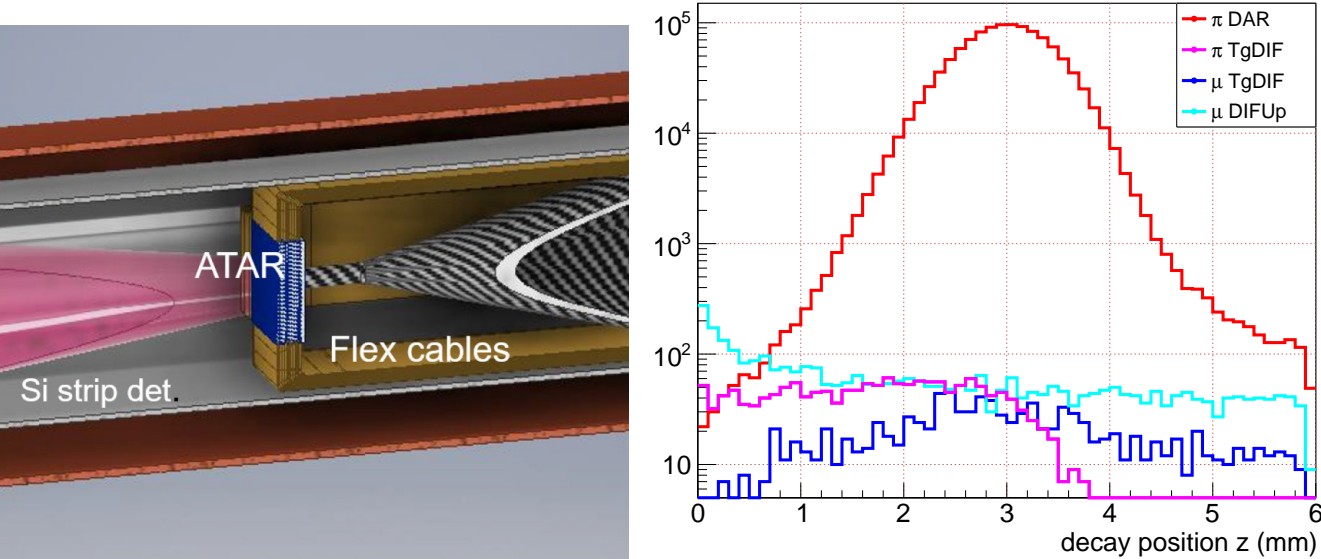

**Figure 2.** (**Left**) Position of the ATAR on the beam line. (**Right**) Simulated pion and muon decay positions along the beam direction (Z) for decay at rest (DAR), decay in flight in the target (TgDIF), and decay in flight upstream of the target (DIFUp). Decay in flight is a potential source of systematic effects, which are currently under study.

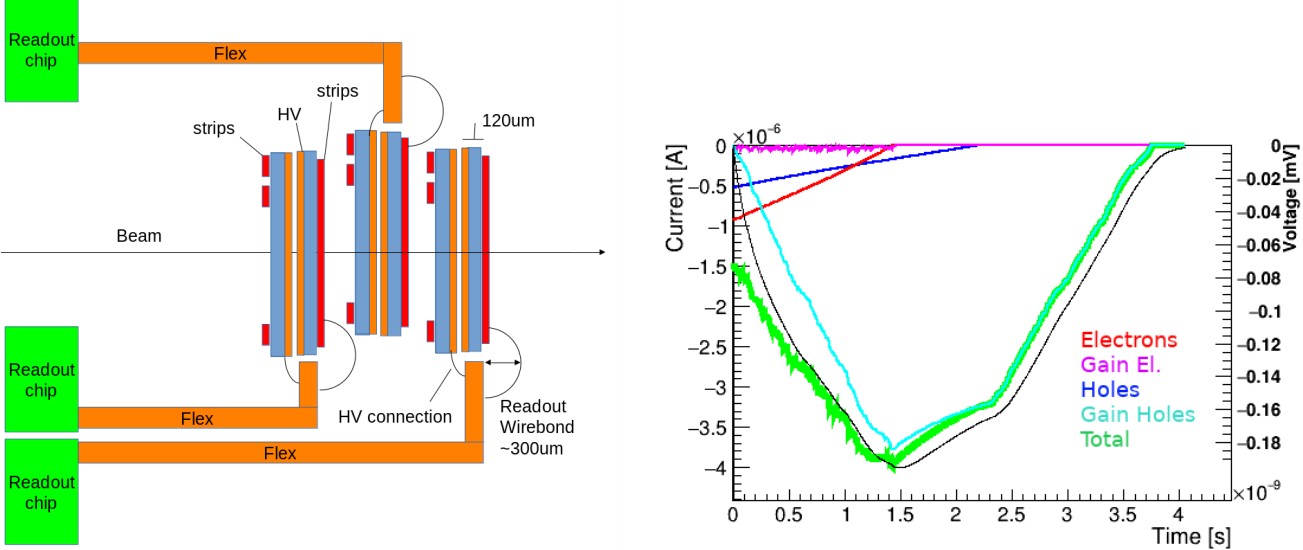

**Figure 3.** (**Left**) Concept schematic design of the ATAR, side view. The flex from the first, third, and fifth sensors is directed in and out of the page. The modules are attached on the HV side and with a few μm of separation on the strip side. The flex to the readout chip is roughly 5 cm long. The total number of sensors is 48, coupled in 24 pairs. (**Right**) Simulated 120 μm thick LGAD pulse shape for a MiP (weightfield2 [25]). The black line is the response of a 2 GHz bandwidth electronic readout.

The foreseen sensor geometry has a pitch of 200 μm with 100 strips mated to a chip with 100 channels and 2 cm width, a standard dimension for microchips. The technology available at the moment for standard LGADs does not allow to have a 200 μm pitch sensor with a fill factor over 80% ([26], Section 5.5.6), it is necessary to exploit one of the new rising LGAD technologies such as AC-LGADs [27], Trench Insulated LGADs [28] or Deep-Junction LGADs [29]. These new technologies were not exploited in a working application yet and require further R&D to maximize their potential. The metal size for the 200-micron pitch strips, as well as other parameters of interest for the sensor such as the doping profile, needs to be chosen after a testing R&D campaign and TCAD Sentaurus or TCAD Silvaco simulations. In addition, some non-typical configurations such as zig-zag strips can be explored to identify muons traveling parallel to a strip.

A best estimate at present for the sensor thickness is around 120 μm to avoid support structures for the sensor, which would introduce dead area. Using fast electronics would result in a pulse with a rise time of about 1 ns for an LGAD of this thickness, as shown in Figure 3 (Right). Such a sensor would be able to separate a single hit from two overlapping hits if they arrive more than 1.5 ns apart. The time resolution on the rising edge should be about 100 ps for the minimum ionizing signals to much better time resolution for large $\pi/\mu$ signals. The exact sensor performance will need to be established by tests done on more mature prototypes.

## 4. LGAD Technology

The chosen technology for the ATAR is LGADs [30], thin silicon detectors with moderate internal gain. LGADs are composed by a low doped region called "bulk" typically 50 μm thick, and a highly doped thin region, a few μm from the electrodes, called 'gain layer'. The electric field in the gain layer is high enough to generate charge multiplication from electrons but not from holes, this mechanism allows having moderate charge multiplication (up to 50) without generating an avalanche. LGADs have a fast rise time and short full charge collection time. However, current standard LGADs have a limitation in terms of granularity and active area. A typical inter-pad gap for a regular LGAD is around 50 to 100 μm, this gap is caused by a protection structure (junction termination extension) at the edge of the high field area of the gain layer to avoid breakdown. To achieve a 100% active area several technologies still at prototype level are being evaluated for PIONEER, such as AC-LGADs [27], TI-LGADs [28], and DJ-LGAD [29].

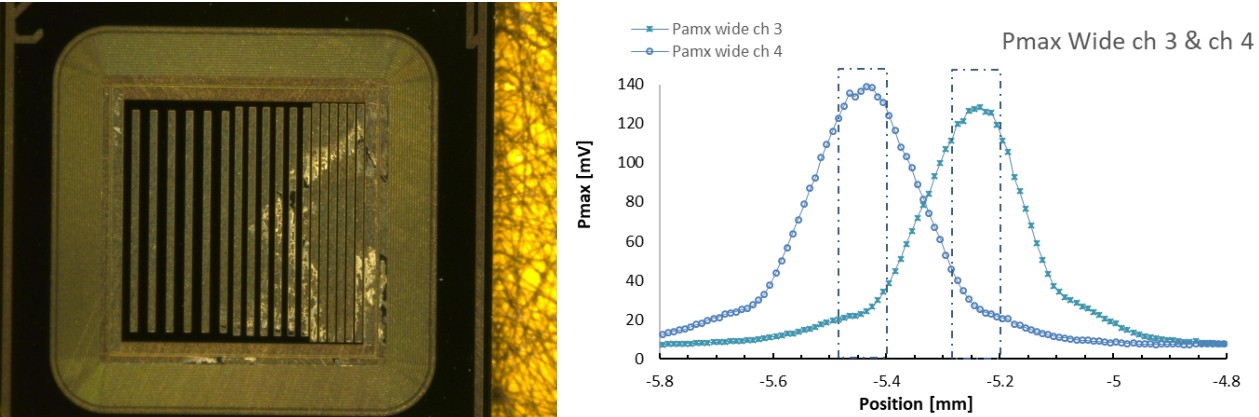

**Figure 4.** (**Left**): Prototype BNL AC-LGAD strip sensor. (**Right**): response (pulse maximum) of two strips as a function of position (perpendicular to the strip) of a sensor with 200 μm of pitch [31]. The dashed lines highlight the position of the two strips in the plot. Data taken at the FNAL test beam facility.

The following studies were made on strip AC-LGAD prototypes from BNL (Figure 4, Left) of which several prototypes were already produced for other projects. AC-LGADs overcome the granularity limitation of traditional LGADs and have been shown to provide a spatial resolution of the order of 10s of μm [32]. This remarkable feature is achieved

with an un-segmented (p-type) gain layer and a resistive (n-type) N-layer. An insulating di-electric layer separates the metal readout pads from the N+ resistive layer. This design also allows having a completely active sensor with no dead regions.

AC-LGADs have intrinsic charge sharing between AC metal pads, so the signal can be picked up by multiple channels at the same time. In a low-density environment, such as PIONEER, this allows having a sparse electrode distribution but with elevated position resolution. Charge depositions can be then reconstructed by calculating the fraction of signal present in all nearby electrodes. For a strip geometry, the hit reconstruction can be made just by looking at the charge fraction between two neighboring strips. The position resolution perpendicular to the strips for the studied 200 μm pitch is <10 μm across the sensor, a few % of the pitch. One strip was tested with readout connected at both ends, for this particular configuration the signal is additionally split between the two ends of the strip. By applying the same fractional method it is possible to reconstruct the hit position also in the direction parallel to the strip. A precision of a few hundred μm was found with 2 mm strips, giving a position resolution of 10% of the strip length.

The sensors have been tested with a laboratory IR laser TCT station [33] and at an FNAL test beam [31]. In both setups the sensors are mounted on fast analog amplifier boards (16 channels) with 1 Ghz of bandwidth (designed at FNAL), the board is read out by a fast oscilloscope (2 GHz, 20 Gs). The response of two strips of a 200 μm pitch BNL AC-LGAD as a function of position can be seen in Figure 4 (Right).

AC-LGADs have several parameters that can be tuned to optimize the sensor response to the specific application. The geometry of the electrodes in terms of pitch and pad dimension is the most important one, however also the N+ resistivity and the insulator thickness between N+ and electrodes influence the charge sharing mechanism. These parameters have been studied with the TCAD Silvaco [34] simulation software (Figure 5) to have a good representation of the observed sensor performance. The calibrated simulation will be used as input to prototype productions to optimize the sensor design for the PIONEER application.

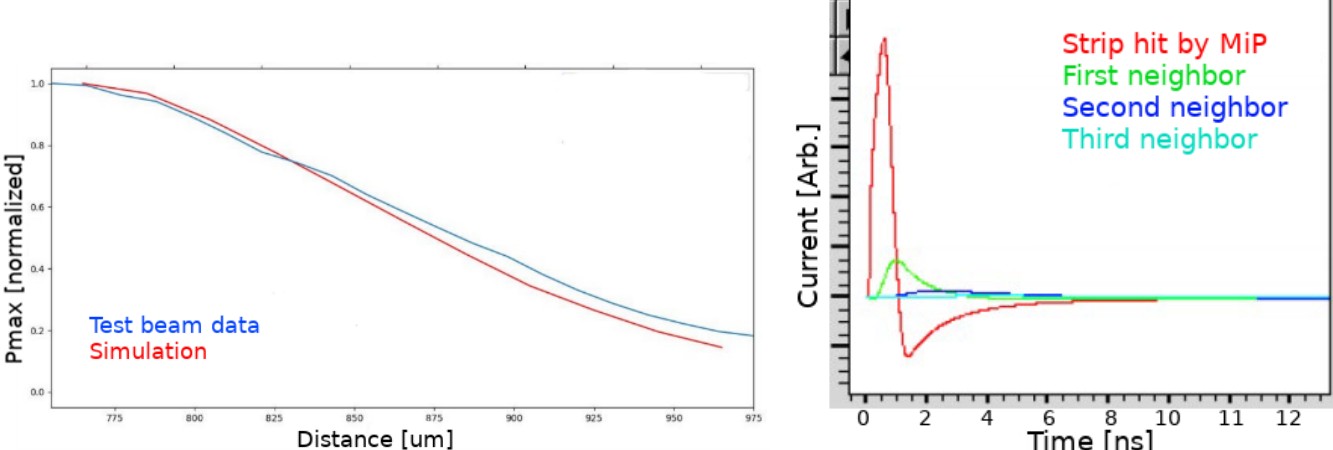

**Figure 5.** (**Left**) Pulse maximum as a function of position for a single strip 80 μm wide in a sensor with a pitch 200 μm, left edge of the plot is the center of the strip. Curves are for data (FNAL testbeam) in blue and TCAD Silvaco simulation in red. (**Right**) AC-LGAD strip simulated waveform with TCAD Silvaco, charge deposition is the strip with the red waveform, green is first neighbor, blue is second neighbor.

As already, stated a dynamic range from MiP (positron) to several MeV (pion/muon) of deposited charge is expected. For this reason, the response of prototype sensors to high charge deposition have to be studied. High ionizing events might affect the charge sharing mechanism, inducing charge also in electrodes far away from the charge deposition. Since the event reconstruction relies on temporal pulse separation the response to successive MiP and high charge deposition have to be studied [35]. Furthermore the effect of gain

suppression for large charge deposition in LGADs has to be taken into account [36]. To study these effect the aforementioned sensors will be tested next year at the ion beam line of the University of Washington (CENPA) to study the response to high ionizing events.

## 5. Electronics and Readout Chain

To readout the ATAR sensors, two crucial electronic components need to be developed: an amplifier chip and a digitizer board. The first chip sits on a board, separated from the sensor and connected through a flex. The amplifier needs to have enough bandwidth for the 120 μm thick sensor in use, 1 GHz should be sufficient. However, the high dynamic range (2000) requirement for the ATAR brings major complications to the readout. Current fast readout chips usually have a dynamic range that is <1000 since they are targeted to MiPs that only detect the tracker sub-system. One possibility is to develop an amplifier chip with a logarithmic response as well as a high enough bandwidth, currently no such chip exists with the necessary characteristics. Another possibility is to stream to the digitizer both the amplified (for MiP) and non-amplified (non MiP) signals or use two different amplifiers with different gains. Yet another option is to adopt a chip capable of dynamic gain switching. Nevertheless, already available integrated chips, such as FAST [37] and FAST2, will be evaluated.

To successfully reconstruct the decay chains, the ATAR is expected to be fully digitized at each event. The fast charge collection time of thin LGADs will allow separating subsequent charge depositions by using advanced deconvolution algorithms. To achieve this goal a high bandwidth digitizer with sufficient bandwidth and sampling rate have to be identified. The same issue afflicting the amplifier, the high dynamic range, is also problematic for the digitization stage. A digitizer that would suit PIONEER needs to be identified, a ready commercial solution would be the best option but the cost per channel might be prohibitive. For this reason, the collaboration is exploring the possibility to develop a new kind of digitizer specific to this application.

Since the amplifier chip has to be positioned away from the active region, the effect of placing a short (5 cm) flex cable between the sensor and the amplification stage has to be studied. The high S/N provided by LGADs would allow the signal to travel without compromising the transmitted information. A prototype flex is being developed and the effect on LGAD signals will be studied.

The DAQ system will utilize FPGA-based fast triggers from the active target and the electromagnetic calorimeter, a trigger correlator that monitors the ATAR / CALO triggers and defines the sparsification and initiates the readout, and a synchronous control system to manage the data transfer of ATAR / CALO data to frontend CPUs. The trigger correlator processes the ATAR topology triggers and CALO channel triggers to identify the various "physics" triggers, control the data sparsification, and initiate the data readout.

## 6. Conclusions

The PIONEER experiment will provide the best measurement up to date for $R_{e/\mu}$ and pion beta decay, improving by an order of magnitude the measurements of PIENU, PEN, and PIBETA. To achieve this goal PIONEER will feature a high granularity calorimeter with high energy resolution and a full silicon active target. The ATAR will allow reconstructing, event by event, the pion decay processes thanks to the high granularity and fast collection time. The technology that was chosen for the ATAR is a high granularity LGAD prototype such as AC-LGADs. Several challenges will be faced for the ATAR design: high dynamic range, clear temporal pulse separation, full digitization, and necessity of low dead material all around.

**Funding:** This paper was prepared for the Emerging PIONEER Collaboration. This work was supported by the United States Department of Energy, grant DE-FG02-04ER41286, and partially performed within the CERN RD50 collaboration.

**Institutional Review Board Statement:** Not applicable.

**Informed Consent Statement:** Not applicable.

**Data Availability Statement:** Not applicable.

**Conflicts of Interest:** The author declares no conflict of interest.

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
