# Peer review of "An LGAD-Based Full Active Target for the PIONEER Experiment"

_instruments, doi:10.3390/instruments5040040_

Round 1

Reviewer 1 Report

The paper "An LGAD-based full active target for the PIONEER experiment" provides a nice introduction to the newly proposed PIONEER experiments and insights on the chosen technology. This is the main message that the reader gets from this paper.

-Has a similar flex technology been tried before? If yes, comment and give references in the paper

-Page 4: "fill factor over 80%", give reference if available

-Page 4: justification of the technology. Brief comment why LGAD technology, despite the need of cables,  has been preferred to monolithic pixel detector with fast timing (for example FASER has a development that aims at the same time direction, although the final chip is tailored to their needs.  There must also be more monolithic developments). 

- Figure 3 Left: give an estimate of the distance between modules and how many stations (3x2 stations from the plot)

Page 5, Figure 4 left. The table on the picture does not seem to mean anything. 

Page 5: there are no analysis plots done with the laser?

Figure 5 Caption: "80 um size and 200 um strip", I cannot understand what it means

Page 6, line 6, N+ has never been defined 

Page 6: Reference TCAD Silvaco

Paragraph 5, page 6, last lines. One other possibility to overcome the dynamic range limitation is a during acquisition dynamic gain switching, like the one implemented for Free Electron Lasers readout chips, check AGIPD for example.  

English or minor editing (not a comprehensive list):

-Author: "PIONEER emerging" -> I would have thought "emerging PIONEER"

- LGAD is defined in the first line of paragraph 4, page 5, but used first in paragraph 3. Define there.

Author Response

Reviewer #1

The paper "An LGAD-based full active target for the PIONEER experiment" provides a nice introduction to the newly proposed PIONEER experiments and insights on the chosen technology. This is the main message that the reader gets from this paper.

> Thank you for your comments, the response are following in-line

-Has a similar flex technology been tried before? If yes, comment and give references in the paper
> This approach hasn't been tried before, it is in our top priority to test a sensor prototype with flex before amplification. A flex prototype has been ordered and will be tested in the next months. We believe the internal gain of LGADs will allow this kind of data transmission.

-Page 4: "fill factor over 80%", give reference if available
> added reference to HGTD TDR section on the TCT measurements of IP gaps

-Page 4: justification of the technology. Brief comment why LGAD technology, despite the need of cables,  has been preferred to monolithic pixel detector with fast timing (for example FASER has a development that aims at the same time direction, although the final chip is tailored to their needs.  There must also be more monolithic developments). 
> LGADs are preferable because monolithic pixels usually have non depleted regions between pixels, furthermore to achieve the same level of time resolution the power needs to be increased to a level that is not sustainable for the ATAR. If the amplifier chip is removed from the compact region of the ATAR the heat dissipation is less of an issue. 

- Figure 3 Left: give an estimate of the distance between modules and how many stations (3x2 stations from the plot)
> Added the information in the caption.

Page 5, Figure 4 left. The table on the picture does not seem to mean anything. 
> it's a table of the distances shown in the picture, it's not readable so it was switched with a picture without the table

Page 5: there are no analysis plots done with the laser?
> It was originally planned to show studies with the laser, but eventually we decided to only show the response using the FNAL test beam data. The text has been adjusted now.

Figure 5 Caption: "80 um size and 200 um strip", I cannot understand what it means
> Added a clarification in the caption

Page 6, line 6, N+ has never been defined 
> Added a brief description of AC-LGAD design on page 5

Page 6: Reference TCAD Silvaco
> added

Paragraph 5, page 6, last lines. One other possibility to overcome the dynamic range limitation is a during acquisition dynamic gain switching, like the one implemented for Free Electron Lasers readout chips, check AGIPD for example.  
> Thank you for the suggestion, dynamic gain switching is a possibility that we're looking into. I added a sentence on it.

English or minor editing (not a comprehensive list):

-Author: "PIONEER emerging" -> I would have thought "emerging PIONEER"
> fixed

- LGAD is defined in the first line of paragraph 4, page 5, but used first in paragraph 3. Define there.
> fixed

Reviewer 2 Report

The paper briefly describes the PIONEER experiment and the chosen technology for the high granularity active target sub-detector. The manuscript is quite well structured, but some modifications are still necessary. So, I ask for the following revision before the paper can be considered for publication. Please indicate the line numbers in the next draft. This will help in pointing comments/corrections if any.

Page 2:

- Fig 1: Improve the quality of the right sketch

- add a reference for Liquid xenon and LSO crystals option.

Page 4, it is recommended a review of the description of the detector using shorter sentences . Some punctual modifications are proposed below:

- In the design the strips would be wire bonded to a flex  —> strips are wire bonded

- This would bring —> This brings

- electronic would sit —> electronics sit

- would be bring the amplified signal —> brings the amplified signal 

- PCb —> PCB

- back end —> back-end

- Figure 3: Improve the  legend readability of the right plot

- so that a sensors would have 100 mated —> with 100 strips mated

Page 5:

- add a reference for the TCT

Page 6:

- The the calibration —> The calibration

- Figure 5: both plots are of poor quality, please improve them

- Amplification --> amplifier 

- Digitisation —> digitiser 

- Rephrase properly the following sentence ”The chip needs to be …….. 1 GHz should be sufficient”

Page 7:

- In "fully digitised at each event, the fast charge" sentence please replace the comma with a point 

- PIONEERS’s needs needs to be —> PIONEER’s needs to be 

- Amplification chip —> Amplifier chip

Reference: please unify the way you report them according to the journal requirements. In most of the references there are missing spaces between author and journal names, please correct it.

Author Response

Reviewer #2

The paper briefly describes the PIONEER experiment and the chosen technology for the high granularity active target sub-detector. The manuscript is quite well structured, but some modifications are still necessary. So, I ask for the following revision before the paper can be considered for publication. Please indicate the line numbers in the next draft. This will help in pointing comments/corrections if any.

> Thank you for your comments, the response are in-line. In the next version I'll set the line numbers.

Page 2:

- Fig 1: Improve the quality of the right sketch
> Improved

- add a reference for Liquid xenon and LSO crystals option.
> added a reference

Page 4, it is recommended a review of the description of the detector using shorter sentences . Some punctual modifications are proposed below:

- In the design the strips would be wire bonded to a flex  —> strips are wire bonded
> fixed
- This would bring —> This brings
> fixed
- electronic would sit —> electronics sit
> fixed
- would be bring the amplified signal —> brings the amplified signal 
> fixed
- PCb —> PCB
> fixed
- back end —> back-end
> fixed

- Figure 3: Improve the  legend readability of the right plot
> Improved

- so that a sensors would have 100 mated —> with 100 strips mated
> fixed

Page 5:

- add a reference for the TCT
> added

Page 6:

- The the calibration —> The calibration
> fixed
- Figure 5: both plots are of poor quality, please improve them
> The quality has been improved
- Amplification --> amplifier 
> fixed
- Digitisation —> digitiser 
> fixed
- Rephrase properly the following sentence ”The chip needs to be …….. 1 GHz should be sufficient”
> fixed

Page 7:

- In "fully digitised at each event, the fast charge" sentence please replace the comma with a point 
> fixed
- PIONEERS’s needs needs to be —> PIONEER’s needs to be 
> fixed
- Amplification chip —> Amplifier chip
> fixed

Reference: please unify the way you report them according to the journal requirements. In most of the references there are missing spaces between author and journal names, please correct it.
> Reference should be correct now